# BPCNN: Bi-Point Input for Convolutional Neural Networks in Speaker Spoofing Detection

**DOI:** 10.3390/s22124483

**Published:** 2022-06-14

**Authors:** Sunghyun Yoon, Ha-Jin Yu

**Affiliations:** 1Department of Artificial Intelligence, Kongju National University, Cheonan 31080, Korea; syoon@kongju.ac.kr; 2School of Computer Science, University of Seoul, Seoul 02504, Korea

**Keywords:** bidirectional feature segmentation, bi-point input, convolutional neural network (CNN), spoofing detection, variable-length features

## Abstract

We propose a method, called bi-point input, for convolutional neural networks (CNNs) that handle variable-length input features (e.g., speech utterances). Feeding input features into a CNN in a mini-batch unit requires that all features in each mini-batch have the same shape. A set of variable-length features cannot be directly fed into a CNN because they commonly have different lengths. Feature segmentation is a dominant method for CNNs to handle variable-length features, where each feature is decomposed into fixed-length segments. A CNN receives one segment as an input at one time. However, a CNN can consider only the information of one segment at one time, not the entire feature. This drawback limits the amount of information available at one time and consequently results in suboptimal solutions. Our proposed method alleviates this problem by increasing the amount of information available at one time. With the proposed method, a CNN receives a pair of two segments obtained from a feature as an input at one time. Each of the two segments generally covers different time ranges and therefore has different information. We also propose various combination methods and provide a rough guidance to set a proper segment length without evaluation. We evaluate the proposed method on the spoofing detection tasks using the ASVspoof 2019 database under various conditions. The experimental results reveal that the proposed method reduces the relative equal error rate (EER) by approximately 17.2% and 43.8% on average for the logical access (LA) and physical access (PA) tasks, respectively.

## 1. Introduction

Automatic speaker verification (ASV) is a technique to verify a user’s identity from his or her utterances. It is convenient and has provided consistently low verification errors. However, ASV is vulnerable to various spoofing attacks that attempt to deceive the system and manipulate its results using a fake utterance. A spoofed utterance is one artificially manipulated to sound like the target speaker’s utterance. Current spoofing attacks include text-to-speech (TTS) synthesis, voice conversion (VC), and replay attacks. Although the identity claim with spoofed utterance must be rejected for obviating the harms caused by the impersonation, ASV may not be able to reject these claims because it is hard to distinguish between a human being’s genuine utterance and the spoofed utterance.

The development of spoofing detection systems to address the shortcomings of ASV has recently gained importance. Spoofing detection is a technique to protect ASV systems from spoofing attacks by distinguishing between genuine and spoofed utterances. Spoofing detection systems reject the identity claim with a spoofed utterance, regardless of how similar the spoofed utterance is to the target speaker’s genuine utterance. For research and development on spoofing detection techniques, ASV spoofing and countermeasure (ASVspoof) challenges have been organized periodically [1,2,3,4]. In this paper, we focus on ASVspoof 2019 [3], which first addresses all major types of spoofing attacks (i.e., TTS, VC, and replay attack).

It is essential for spoofing detection to capture the differences of the frequency attributes between genuine and spoofed utterances, regardless of the type of spoofing attack. The frequency attributes of genuine and spoofed utterances differ [5,6]. Furthermore, the discriminative attributes between genuine and spoofed utterances are spread across the entire time domain of an utterance and there is no strong time dependency. This locally connected information can be effectively modeled by convolutional neural networks (CNNs) [7]. Accordingly, many cutting-edge spoofing detection systems are based on CNNs [8,9,10,11].

Like other types of neural networks, a CNN is also trained in a mini-batch unit [12]. When feeding input features into a CNN in a mini-batch unit, all features in each mini-batch must have the same size. However, the lengths of speech utterances differ because speakers have different uttering styles, even if all speakers are constrained to utter only the same phrase. Therefore, it is required that all input utterances (it is not limited to ‘utterance’ and can correspond to any kind of sequence (e.g., a waveform signal, a sequence of acoustic feature vectors), although we use the word ‘utterance’ in this sentence) have the same length before we feed them into a CNN. There are several ways to make all input features in each mini-batch have the same length, which can be categorized into three methods: (1) padding only, (2) both padding and truncation, and (3) feature segmentation. Throughout this paper, the *target length* refers to the length that all features in a mini-batch must have.

The first method is to apply padding only, by repeating the feature’s own frames or filling with zeros (i.e., zero-padding). In other words, there is no feature truncated. The target length is set long enough (e.g., the length of the longest feature over the dataset) so that no feature is longer than the target length, as in [13,14,15]. This method has the advantage of no information loss caused by truncation. However, it causes a data redundancy problem caused by padding because the padded frames are not novel but merely part of the original frames (for the repeating) or common zero values that are meaningless (for the zero-padding). Thus, padding brings the additional unnecessary information in terms of the information diversity and novelty, which may not help improve classification. Furthermore, this problem accompanies the increases in the computational burden and memory usage. This side effect worsens as the standard deviation of the utterance lengths increases.

The second method is to apply both padding and truncation. The target length is set to a moderate length, neither too long nor too short (e.g., the average length), as in [9,10,15,16,17,18,19,20,21,22,23,24]. Features shorter than the target length are padded and those longer than the target length are truncated. This method has a relatively smaller computational burden and memory usage than the first method. However, it causes the information distortion that is due to both padding and truncation. The padding leads to the additional unnecessary information, as described above, and truncation leads to the information loss because of not being able to use truncated frames.

In [25], an on-the-fly data loader was proposed to handle variable-length input features. It can be regarded as an extension of the aforementioned second method to enable batch-wise variable-length training. For each iteration, the target length (common to all the features in each mini-batch) and start points (for truncation only; different for each input feature in a mini-batch) are randomly selected. This method achieved improved performances on both text-independent speaker verification and language identification [25]. However, our findings reveal that the on-the-fly data loader was shown to be ineffective for improving performance on spoofing detection, except in some cases. This issue will be discussed in Section 5.2.

The third method is feature segmentation [8,11,26,27,28,29], which breaks each feature down into fixed-length segments. The target length corresponds to the length of a segment. A CNN receives one segment at one time rather than an entire feature. This method seems to combine the benefits of the previous two methods. It can reduce the information distortion caused by padding and truncation. Therefore, it is a more flexible method in terms of the difference in the utterance length. In this paper, we used the feature segmentation as a baseline method.

However, segmentation has the shortcoming of the limitation on the amount of information available at one time because a CNN can consider only one segment at one time, not an entire feature. At least, but not limited in spoofing detection, this shortcoming is detrimental to the performance. It is important for better detection to capture and model as many the discriminative attributes in an input feature as possible, and then aggregate them into a fixed-size embedding. Moreover, the discriminative attributes are spread across the entire time domain, rather than being concentrated in certain time intervals, as mentioned earlier. In other words, the feature segmentation is inevitable to reach suboptimal performances on spoofing detection. This problem has made us take how to increase the amount of information available at one time into consideration.

We had proposed a method called multiple points input for CNN [27]. The proposed method increases the amount of information available at one time by feeding a pair of two segments as an input to a CNN. It exhibited substantial reductions in detection error in the ASVspoof 2019 PA scenario (i.e., replay attack detection task). Nonetheless, there are still some issues worthy of further discussions in [27] because our previous experiments in [27] were performed under restricted conditions: (1) the use of only one type of CNN-based model, squeeze-and-excitation [30] residual network [31] (SE-ResNet), (2) a fixed target length at 400 frames, and (3) evaluation only in the replay attack detection task. In this paper, therefore, we present comprehensive experimental results conducted under various conditions and in-depth analyses to verify the ability of the proposed method to reduce the detection errors in various conditions. The contributions of this paper are summarized as follows:Proposing a bi-point input for CNN (BPCNN). It feeds a pair of two segments, rather than one segment, into a CNN at one time. The main advantage of BPCNN is to increase the amount of the available information at one time with hardly changing the CNN structure.Proposing various methods for combining the two segments in various levels: embedding-level combination, feature map-level combination, and two-channel input.Evaluating the performances of the proposed method in both logical and physical access tasks in various conditions and analyzing the effect and strength of the proposed method via the ablation studies.

The structure of this paper is organized as follows. Section 2 outlines the conventional feature segmentation method. Section 3 introduces the proposed method. Section 4 describes the experimental setup, and Section 5 analyzes the obtained results. Finally, Section 6 concludes the paper.

## 2. Conventional Feature Segmentation

Feature segmentation is a method that decomposes all the variable-length features into fixed-length segments along the time axis. Figure 1 and Algorithm 1 illustrate the overview and pseudocode of conventional feature segmentation. Let $X=\left[x_{0},\ldots ,x_{T-1}\right]$ be a sequence of T feature frames extracted from an utterance (e.g., log power spectrogram). With the feature segmentation, X is divided into N segments ${\left\{F_{i}\right\}}_{i=0}^{N-1}$ using a sliding window of M(<T) frames length with L frames shift (this explanation is only for the case of M<T. The case of M≥T will be discussed later.) All segments have the same length of M, which is the target length. The number of segments N is different for each feature, which is proportional to the feature length T. The i-th segment $F_{i}$ corresponds to the sequence of M frames that start from the iL-th frame $x_{iL}$ to the (iL+M−1)-th frame $x_{iL+M-1}$, $F_{i}=\left[x_{iL},\;\ldots ,\;x_{iL+M-1}\right]$, where iL+M≤T for all i.

**Algorithm 1** Pseudocode of Conventional Feature SegmentationInput  $X=\left[x_{0},\;\ldots ,\;x_{T-1}\right]$: A feature with the length of T  M: The segment length  L: The shift interval for segmentationOutput  S: A list of the segments1.**if** T>M:
2.  
S=[ ]
# Empty list3.  
N=⌊MT⌋
# The quotient4.  
r=mod(T−M,L)
# The remainder5.  **for** i=0 to N−1:
6.    
$F_{i}=\left[x_{iL},\;\ldots ,\;x_{iL+M-1}\right]$
# The i-th segment7.    
$S.\mathrm{append}\left(F_{i}\right)$

8.  **if** r>0:
9.    
$F_{N}=\left[x_{T-M},\;\ldots ,\;x_{T-1}\right]$
# The last M frames of X10.    
$S.\mathrm{append}\left(F_{N}\right)$

11.  **return** S
12.**else if** T<M:
13.  
$F_{0}=\mathrm{pad}\left(X,M\right)$
# Set X to have the length of M14.  
**return**
$\left\{F_{0}\right\}$

15.**else**:# The case of T==M16.  
$F_{0}=X$
# X itself becomes a segment17.  
**return**
$\left\{F_{0}\right\}$



We do not consider a unified feature map approach [11,16,22] for segmentation. In the unified feature map approach, all features are first extended to have the same length that are sufficiently long (e.g., usually the length of the longest feature over the dataset) before segmentation. Therefore, all features have the same number of segments regardless of their original lengths. However, we empirically found no significant difference in detection error between with and without the unified feature map. Nonetheless, our approach (i.e., without the unified feature map) is more efficient in terms of data redundancy and computational amount and memory usage. This is why the unified feature map approach is not considered throughout this paper.

The following two issues should be considered to implement the feature segmentation. The first issue is how to handle the features shorter than or equal to the target length (i.e., M≥T), where segmentation cannot be applied. As described above, M should be set to a moderate length, neither too long nor too short. In most cases, therefore, there are some features shorter than M. In our approach, we extend the features shorter than M to have the length of M by padding M−T frames from those features. If the number of additionally required frames M−T is below the target length T (i.e., M−T<T), a padded segment $F_{0}$ is obtained by padding the first M−T frames of X at the end of X, $F_{0}=\mathrm{pad}\left(X,M\right)=\left[x_{0},\ldots ,x_{T-1},x_{0},\ldots ,x_{M-T-1}\right]=\left[X,x_{0},\ldots ,x_{M-T-1}\right]$. Otherwise (i.e., M−T≥T), $F_{0}$ is defined as $Q=\lfloor \frac{M}{T}\rfloor$ times repeat of X followed by the first M−QT frames of X, $F_{0}=\mathrm{pad}\left(X,M\right)=\left[X,\ldots ,X,x_{0},\ldots ,x_{M-QT-1}\right]$. When M=T, $F_{0}$ just corresponds to X itself. In turn, there is only one segment $F_{0}$ for the feature whose length is shorter than or equal to M.

The second issue is how to handle the remaining frames fewer than M after segmentation. For each feature of which the length is T(>M), there are R=mod(T−M,L) frames left after segmentation, where mod(a,b) is the modulo operation with the dividend a and divisor b. We also use the remaining R frames to exploit complete frames. When R>0, take an additional segment corresponding to the last M frames of X, regardless of the number of remainder frames R. This segment is defined as the last segment $F_{N}=\left[x_{T-M},\;\ldots ,\;x_{T-1}\right]$.

In the training phase of the proposed method, each segment is treated as individual data, regardless of which feature the segment was obtained from. Even so, all segments obtained from a feature share the same class label. For instance, when the class label of a feature X is c, all the segments ${\left\{F_{i}\right\}}_{i=0}^{N-1}$ from X also have the class label c. In the evaluation phase, the score (e.g., log probability of the genuine class) of each feature is calculated by aggregating (e.g., averaging) the scores of all the segments obtained from the feature.

## 3. The Proposed Method

The proposed method alleviates the problem of the conventional method that the available information at one time is limited because CNN takes only one segment $F_{i}$ at one time. The proposed method consists of two steps: (1) bidirectional feature segmentation (cf., unidirectional feature segmentation in the conventional method) and (2) bi-point input (cf., one-point input in the conventional method).

### 3.1. Bidirectional Feature Segmentation

In the conventional feature segmentation described in Section 2, a feature is decomposed along only one direction: the positive time direction. In contrast, in the proposed method, a feature is decomposed along two opposite directions: the positive and negative time directions. It is inspired by the bidirectional recurrent neural network (BRNN) [32]. BRNNs are developed to increase the amount of available information. They can obtain twice as much information as standard RNNs, using the information in both the forward (responsible for the positive time direction) and backward (responsible for the negative time direction) sequences.

Figure 2 and Algorithm 2 illustrate the overview and pseudocode of bidirectional feature segmentation. Given a feature $X=\left[x_{0},\ldots ,x_{T-1}\right]$, bidirectional feature segmentation is processed in two steps: forward and backward steps. The forward step is identical to conventional feature segmentation. In the forward step, X is divided by moving the sliding window, of which the length and shift interval are M(≤T) and L, respectively, along the positive time direction (i.e., from 0 to T−1) to obtain N forward segments ${\left\{F_{i}\right\}}_{i=0}^{N-1}$. The i-th forward segment $F_{i}$ is $\left[x_{iL},\ldots ,\;x_{iL+M-1}\right]$, as explained in Section 2. In contrast, in the backward step, X is divided by moving the sliding window along the negative time direction (i.e., from T−1 to 0). After the backward step is complete, we can obtain N backward segments ${\left\{B_{i}\right\}}_{i=0}^{N-1}$. Let $\tilde{X}=\left[x_{T-1},\ldots ,x_{0}\right]$ be the flipped order of X. The i-th backward segment $B_{i}$ is $\left[x_{T-iL-1},\ldots ,\;x_{T-iL-M}\right]$, where iL+M≤T for all i.

**Algorithm 2** Pseudocode of Bidirectional Feature SegmentationInput  $X=\left[x_{0},\;\ldots ,\;x_{T-1}\right]$: A feature with the length of T  $\tilde{X}=\left[x_{T-1},\;\ldots ,\;x_{0}\right]$: The flipped order of X  M: The segment length  L: The shift interval for segmentationOutput  S: A list of the segment pairs1.**if** T>M:
2.  
S=[ ]
# Empty list3.  
N=⌊MT⌋
# The quotient4.  
r=mod(T−M,L)
# The remainder5.  **for** i=0 to N−1:
6.    
$F_{i}=\left[x_{iL},\ldots ,\;x_{iL+M-1}\right]$
# The i-th forward segment7.    
$B_{i}=\left[x_{T-iL-1},\ldots ,\;x_{T-iL-M}\right]$
# The i-th backward segment8.    
$S.\mathrm{append}\left(F_{i},B_{i}\right)$

9.  **if** r>0:
10.    
$F_{N}=\left[x_{T-M},\;\ldots ,\;x_{T-1}\right]$
# The last M frames of X11.    
$B_{N}=\left[x_{M-1},\ldots ,x_{0}\right]$
# The last M frames of $\tilde{X}$12.    
$S.\mathrm{append}\left(F_{i},B_{i}\right)$

13.  **return** S
14.**else if** T<M:
15.  
$F_{0}=\mathrm{pad}\left(X,M\right)$
# Set X to have the length of M16.  
$B_{0}=\mathrm{pad}\left(\tilde{X},M\right)$
# Set $\tilde{X}$ to have the length of M17.  **return** $\left\{F_{i},B_{i}\right\}$
18.**else**:# The case of T==M19.  
$F_{0}=X$
# X itself becomes a forward segment20.  
$B_{0}=\tilde{X}$
# X itself becomes a backward segment21.  **return** $\left\{F_{i},B_{i}\right\}$


When M>T, one forward segment $F_{0}$ and one backward segment $B_{0}$ can be obtained without segmentation, as in the conventional method. The method of obtaining the forward segment $F_{0}$ is identical to the conventional method. The backward segment $B_{0}$ can be obtained as follows. If M−T<T, where M−T corresponds to the number of additionally required frames, $B_{0}$ is obtained by padding the first M−T frames of $\tilde{X}$ at the end of $\tilde{X}$, $B_{0}=\mathrm{pad}\left(\tilde{X},M\right)=\left[x_{T-1},\ldots ,x_{0},x_{T-1},\ldots ,x_{2T-M}\right]$ (the first M−T frames of $\tilde{X}$ correspond to the last M−T frames of X with the flipped order of frames, $\left[x_{T-1},\ldots ,x_{2T-M}\right]$, where 2T−M=T−(M−T)) Otherwise (i.e., M−T≥T), $B_{0}$ is defined as $Q=\lfloor \frac{M}{T}\rfloor$ times repeat of $\tilde{X}$ followed by the first M−QT frames of $\tilde{X}$, $B_{0}=\mathrm{pad}\left(\tilde{X},M\right)=\left[\tilde{X},\ldots ,\tilde{X},x_{T-1},\ldots ,x_{\left(Q+1\right)T-M}\right]$ (the first M−QT frames of $\tilde{X}$ correspond to the last M−QT frames of X with the flipped order of frames, $\left[x_{T-1},\ldots ,x_{\left(Q+1\right)T-M}\right]$, where (Q+1)T−M=T−(M−QT)) For the case of M=T, $B_{0}$ just corresponds to $\tilde{X}$ itself.

If there are remaining R frames after bidirectional feature segmentation, the last forward segment $F_{N}$ is defined the same as in conventional feature segmentation. Similarly, the last backward segment $B_{N}$ is defined as the last M frames of $\tilde{X}$, $B_{N}=\left[x_{M-1},\ldots ,x_{0}\right]$.

All the other conditions not described are the same as those in conventional feature segmentation. For example, the class label of the feature is also shared for all backward segments.

### 3.2. Bi-Point Input

With bidirectional feature segmentation, we can obtain two sets of segments from an utterance: a set of forward segments ${\left\{F_{i}\right\}}_{i=0}^{N-1}$ and a set of backward segments ${\left\{B_{i}\right\}}_{i=0}^{N-1}$. With bi-point input, the i-th forward segment $F_{i}$ forms a pair with the i-th backward segment $B_{i}$. Therefore, there are N pairs of segments. The CNN receives one pair (i.e., two segments, $F_{i}$ and $B_{i}$) as an input at one time, rather than only one (forward) segment. Therefore, the amount of available information at one time becomes up to double. The amount of available information with the bi-point input is usually less than twice the conventional one-point input method because the time ranges of $F_{i}$ and $B_{i}$ could be partially or completely overlapped for some i (e.g., around the middle of the feature) under our approach. The wider the overlapped range, the less the gain in the increase in the amount of information available at one time.

It is reasonable that each pair consists of $F_{i}$ and $B_{i}$, not $F_{i}$ and the (N−1−i)-th backward segment $B_{N-1-i}$. Recall that the primary purpose of the bi-point input is to increase the amount of information available at one time. If the i-th pair is made up of $F_{i}=\left[x_{iL},\ldots ,\;x_{iL+M-1}\right]$ and $B_{N-1-i}=\left[x_{iL+M-1},\ldots ,\;x_{iL}\right]$ (for a brief explanation, we assume there is no remainder frame after segmentation, that is, R=mod(T−M,L)=0, but even if R>0, the increased amount of information available at one time is negligible and has little effect on reducing the detection error), both segments in each pair (i.e., $F_{i}$ and $B_{N-1-i}$) always cover the exact same time range. The only difference between the two segments is the frame order. It indicates that both segments in each pair always have the same information. In this case, therefore, it is difficult to increase the amount of information available at one time. Even if the detection error is reduced, it may be due to the data augmentation based on flipping the time order of feature, rather than the increase in the available information at one time. We will discuss this issue in detail in Section 5.2.

Recall that $F_{i}$ and $B_{i}$ have the opposite frame order (explained in Section 3.1), which not only avoids meaningless operations but also has the side effect of data augmentation. Although $F_{i}$ and $B_{i}$ in a same pair generally cover the different time ranges, there could be some cases where $F_{i}$ and $B_{i}$ cover the same time range at certain i (e.g., around the middle of the feature), as mentioned above. When $F_{i}$ and $B_{i}$ cover the same time range, their outputs are exactly the same if they have the same frame order. In this case, feeding $F_{i}$ and $B_{i}$ together is merely a redundancy of meaningless operations, which is not the increase in the available information at one time nor data augmentation. This problem can be avoided by making the frame orders of $F_{i}$ and $B_{i}$ opposite, although the available information at one time still cannot be increased. As long as the frame orders are opposite, $F_{i}$ and $B_{i}$ can be treated as different features for CNN, even if they cover exactly the same time range. Notice that the words ‘different features’ do not mean that $F_{i}$ and $B_{i}$ have different information, but mean that they just have (slightly) different output values.

Figure 3 illustrates the overall frameworks of (a) the conventional system (i.e., based on conventional feature segmentation and one-point input) and (b–d) the proposed systems (i.e., based on bidirectional feature segmentation and bi-point input). The differences between the conventional and proposed systems are highlighted in blue shade. In the proposed systems, both segments in a pair are fed into the same CNN, which indicates that the CNN parameters are shared across both segments. The results from each of the segments in a pair are combined before being fed into the classifier. From a pair of two segments, therefore, we obtain one score value, not two.

We propose several methods to combine the results from the two segments in a pair, including the methods used in [8,27,29]. These are designed considering that most CNN-based models are composed in the following order of modules: CNN, a global average pooling (GAP) layer, and a classifier. The GAP layer transforms C feature maps of size (C×H×W) into a vector of size C by averaging along each channel, where C is the number of channels. The methods are categorized into four methods, depending on where the combination is performed within the networks.

#### 3.2.1. Embedding-Level Combination

Figure 3b illustrates the embedding-level combination method. We used the output of GAP layer as embedding. In this method, two embeddings obtained from each pair of segments are combined to form one embedding. The combined embedding is then fed into the classifier. We combine the two embeddings with three approaches: concatenating (denoted as *concat*), element-wise maximum (denoted as *vmax*), and element-wise averaging (denoted as *vmean*). Let $u={\left[u_{1},\;\ldots ,\;u_{D}\right]}^{T}$ and $v={\left[v_{1},\;\ldots ,\;v_{D}\right]}^{T}$ be two embedding vectors, and $w={\left[w_{1},\;\ldots ,\;w_{D}\right]}^{T}$ be the output. *vmax* computes the element-wise maximum, so the output w becomes w=max(u,v), where $w_{i}=\max\left(u_{i},v_{i}\right)$. *vmean* computes the element-wise average, so the output w becomes w=0.5(u+v) (i.e., just vector addition followed by scalar multiplication), where $w_{i}=0.5\left(u_{i}+v_{i}\right)$. The dimensionality of the combined embedding is doubled when using *concat*, indicating that the number of parameters for the classifier has doubled. In contrast, when using *vmax* or *vmean*, the combined embedding has the same dimensionality as each original embedding, so no additional parameter is required.

#### 3.2.2. Feature Map-Level Combination

Figure 3c illustrates the feature-map-level combination method. The output of the CNN module with the shape of (C×H×W) corresponds to a set of C feature maps, where C is the number of output channels of the last convolution layer in the module, and H and W are the height and width of each feature map, respectively. Two sets of feature maps can be obtained from two segments in a pair. We combine these two feature maps using an element-wise maximum (denoted as *fmax*). We do not consider other operations for combination, such as concatenating and element-wise averaging per channel, because concatenating and element-wise averaging at the feature map level produce the same results as those at the embedding level (i.e., *concat* and *vmean*, respectively).

#### 3.2.3. Two-Channel Input

Figure 3d illustrates the two-channel input method. Most acoustic features from utterances in the form of spectrograms, corresponding to a CNN input in many cases, are in 2D matrix form with the shape of (H×W). Such a 2D matrix can be viewed as a 3D tensor with the shape of (1×H×W), where the number of channels is 1. We propose the two-channel input method (denoted as *2ch*), where two segments in a pair are concatenated along the channel axis to form one 3D tensor with the shape of (2×H×W). This concatenated 3D tensor is then used as a CNN input. For receiving an input with two channels, the number of kernels of the first convolution layer of the CNN module must be doubled. Consequently, the number of parameters for the first convolution layer doubles.

The CNN designed for one-channel inputs can also receive two-channel inputs without increasing the number of kernels by sharing the kernels across all input channels (denoted as *2ch_s*). In this method, therefore, the number of parameters is not changed. However, we do not consider the *2ch_s* method because of its lower gain in reducing detection error. The comparison of the detection errors of *2ch* and *2ch_s* will be discussed in the last part of Section 5.2.

#### 3.2.4. Statistics-Level Combination

The statistics-level combination method is only for an x-vector network [33,34], one of the models used in our experiments. Because the x-vector network cannot receive inputs with multiple channels, the feature map-level combination and two-channel input methods cannot be used with the x-vector network.

Accordingly, we propose another statistics-level combination method (denoted as *statc*) that can be used only for the x-vector network. The statistics pooling layer originally computes the statistics (e.g., mean and standard deviation) over all the M output frames from one (forward) segment. In contrast, with the *statc* method, the statistics pooling layer computes the statistics over 2M output frames, of which some M frames are from a forward segment and the other M frames are from a backward segment.

## 4. Experiments

### 4.1. Database

We used the ASVspoof 2019 database [35]. It includes two task scenarios: logical access (LA) and physical access (PA).

The LA scenario addresses the spoofing attacks generated by TTS synthesis and VC. It is divided into the training (2580 genuine and 22,800 spoofed speeches), development (2548 genuine and 22,296 spoofed speeches), and evaluation (7355 genuine and 63,882 spoofed speeches) sets. The utterance length is approximately 3.25 s on average and ranges from 0.47 to 13.19 s with a standard deviation of 1.47. Figure 4a illustrates the histogram of the utterance lengths in the LA scenario.

The PA scenario addresses the spoofing attacks generated by replay attack methods. It is divided into the training (5400 genuine and 48,600 spoofed speeches), development (5400 genuine and 24,300 spoofed speeches), and evaluation (18,090 genuine and 116,640 spoofed speeches) sets. The utterance length is approximately 4.28 s on average and ranges from 1.46 to 10.32 s with a standard deviation of 1.2. Figure 4b illustrates the histogram of the utterance lengths in the PA scenario.

The training sets were used to build spoofing detection systems. The development sets were used to validate the detection errors of the systems for every epoch. The evaluation sets were used to evaluate the detection error rates of the system.

### 4.2. Experimental Setup

We used a 257-dimensional log power spectrogram as input feature for all types of CNNs used in our experiments. For each utterance, 25 ms frames were extracted at 10 ms intervals. The frame-level preprocessing was performed in the following order: removing the DC offset, pre-emphasis filtering with a coefficient of 0.97, and applying the Hamming window. The number of fast Fourier transform (FFT) points was 512. We performed neither voice activity detection (VAD) [36] nor cepstral mean and variance normalization (CMVN) for the spectrogram. The Kaldi toolkit [37] was used to extract the feature.

For segmentation, we set the length of segment M to 200 (corresponding to 2 s), 400 (corresponding to 4 s), and 600 (corresponding to 6 s). Approximately 81%, 25%, and 5% of utterances in the LA subset are longer than 2, 4, and 6 s, respectively. Approximately 99%, 56%, and 8% of utterances in the PA subset are longer than 2, 4, and 6 s, respectively. The shift interval L was half of M, which is based on the claim in [26] that segmentation with overlap generally outperforms that without overlap (i.e., L=M).

We used six types of CNN-based models to build spoofing detection systems: SE-ResNet, x-vector network, dense convolutional network (DenseNet) [38], MobileNetV2 [39], ShuffleNetV2 [40], and MNASNet [41] designed using an approach of automated mobile neural architecture search (MNAS). All models used in our experiments have a fully connected (FC) softmax classifier at the top. The number of classes is three (i.e., genuine, TTS, and VC) for the LA, and two (i.e., genuine and replayed) for the PA. AMSGrad [42], a variant of the Adam [43] optimizer, was used to minimize cross-entropy loss. The hyper-parameters for the optimizer were as follows: a learning rate of $10^{-3}$, $\beta_{1}=0.9$, $\beta_{2}=0.999$, $\epsilon =10^{-8}$, and a weight decay of $10^{-4}$. All weights were initialized from the He normal distribution [44] and no bias was used. We trained the networks for 100 epochs with a mini-batch size of 64. PyTorch [45] with the torchvision.models package was used to implement the systems.

We used the equal error rate (EER) as the evaluation metric. The lower the EER, the lower the detection error. For the LA, where the number of classes is three, the score of each segment was defined as the log probability of the genuine class $\log\;p_{g}\left(x\right)$, corresponding to the log-softmax output for that class. For the PA, where the number of classes is two, the score of each segment was defined as the log ratio of the spoof probability to the genuine probability, $\log\;p_{g}\left(x\right)-\log\;p_{s}\left(x\right)$. The score of each utterance is computed by averaging the scores of all the segments in the utterance.

## 5. Results

### 5.1. Experimental Results and Discussion

Table 1 presents the EERs of the baseline and proposed systems on the development and evaluation trials of ASVspoof 2019 LA and PA tasks. For each system (corresponding to the combination of *model* and *method*), we selected the one among 100 epochs that exhibited the lowest EER in the development trials. As guidance, the lowest EERs in the evaluation trials (i.e., the oracle EERs) were additionally reported in Appendix A. The *Fusion* method indicates the score-level fusion of all the proposed systems using simply summation. Unless otherwise noted, the explanations are focused on the results of the evaluation trials.

The experimental results in Table 1 indicate that the proposed method outperforms the conventional feature segmentation on spoofing detection, both the LA and PA tasks. In most cases, the proposed method exhibited lower EERs than the baseline with suitable combination methods, regardless of the type of model, segment length M, and task. There is an exceptional case where the proposed method failed to reduce EER for MobileNetV2 with M=600. This result is mainly due to the limitation of the validation (especially for the LA task), rather than the proposed method’s ineffectiveness at reducing the EER. This issue is explained in Appendix A. Our experiments provide evidence that the discriminative attributes between genuine and spoofed utterances are spread across the entire time domain, rather than concentrated primarily in some specific ranges, as stated in Section 1. The performance would not be improved with the proposed method if the discriminative attributes were concentrated in some specific time ranges. However, even if the attributes are spread over the entire time, we could not get lower EERs if the attributes in different time ranges do not have distinct information in terms of the detection. Therefore, the proposed method is efficient for the cases where the distinct information is spread across the entire time domain, such as spoofing detection.

We discuss the results in Table 1 with respect to the segment length M. The longer the M, the lower the EERs in most cases for both baseline and proposed systems. It is consistent with our intuition that the longer segment has more information than the shorter segment (if M becomes even longer (e.g., thousands of frames or more), EER could become saturated, but we do not compare with the cases where M is extremely long because it is rarely considered because of the limited computing power and memory usage). Some exceptional cases were observed for the LA task where the EERs were increased on average, although M lengthened from 400 to 600: with SE-ResNet, DenseNet (only with the proposed methods), and MNASNet. These cases are discussed in Appendix A.

We compared the baseline with a target length $M_{b}$ and the proposed methods with the target length $M_{p}$ shorter than $M_{b}$ (i.e., $M_{p}<M_{b}$). For $M_{b}=400$ and $M_{p}=200$, it could be expected that the proposed systems exhibit similar EERs to the baseline systems, because the total numbers of frames fed at one time are the same. Following the same logic, it also could be expected for $M_{b}=600$ and $M_{p}=400$ that the proposed systems exhibit lower EERs to the baseline systems because the total number of frames fed at one time is greater with the proposed systems than the baseline systems. However, the proposed systems with $M_{p}=200$ generally exhibited EERs higher than those of the baseline systems with $M_{b}=400$. The proposed systems with $M_{p}=400$ exhibited slightly lower but similar levels of EERs as the baseline systems with $M_{b}=600$ on average. Recall that the amount of information with the proposed method is usually less than twice the amount with the baseline method because of the partially overlapped time ranges, as described in Section 3.2. Accordingly, the above results are acceptable and do not undermine our expectation that the longer M, the more available information at one time, and thus the lower the EERs.

The above results illustrated that the effectiveness of the proposed method on reducing EERs depends on M. We empirically found that the proposed method was more effective at reducing EER as M was closer to the average length. For M=200, 33 out of the 35 proposed systems (including *Fusion*) exhibited lower EERs than the baseline in both the LA and PA tasks. The EERs were reduced on average by approximately 17.2% (−12.1% to 44.6%) for the LA task and 35.6% (−36.9% to 73.4%) for the PA task. For M=400, 28 and 33 out of the 35 proposed systems exhibited lower EERs than the baseline in the LA and PA tasks, respectively. The EERs were reduced on average by approximately 13.4% (−43.5% to 41%) for the LA task and 43.8% (−6.5% to 73.3%) for the PA task. When M is longer than the average, the proposed method exhibited a lower gain in reducing EER. For M=600, 18 out of the 35 proposed systems exhibited higher EERs than the baseline for the LA task. The EERs for the LA task increased on average by approximately 2.7% (i.e., decreased approximately −2.7%, range from −74.8% to 25%) when using the proposed method. For the PA task, 27 out of the 35 proposed systems exhibited lower EERs than the baseline. The EERs were reduced on average by approximately 18.2% (−100.5% to 64.4%). The proposed method was most effective for reducing EER when M=200 for the LA task and when M=400 for the PA task. These results can be interpreted as follows. For the features longer than M, the proposed method enables the CNN to consider more information at one time. In contrast, the proposed method has a negligible effect for features shorter than M because the CNN already considers the information of the complete feature at one time for these short features, even without the proposed method. Thus, for the features shorter than M, there is no longer room to increase the amount of information available at one time.

We provide a rough guidance to set a proper M when using the proposed method without evaluation by defining the additional gain α=Mρ, where ρ is the proportion of the number of the features longer than M in the dataset (stated in Section 4.2). For the LA dataset, where the average number of frames is 325, α=200×0.81=162 when M=200 (i.e., 81% of features have frames more than 200), α=400×0.25=100 when M=400 (i.e., 25% of features have frames more than 400), and α=600×0.05=30 when M=600 (i.e., 5% of features have frames more than 600). The average EER reductions are approximately 17.2%, 13.4%, and −2.7% (i.e., the EERs were increased by approximately 2.7% on average) when M=200, M=400, and M=600, respectively. For the PA dataset, where the average number of frames is 428, α=200×0.99=198 when M=200 (i.e., 99% of features have frames more than 200), α=400×0.56=224 when M=400 (i.e., 56% of features have frames more than 400), and α=600×0.08=48 when M=600 (i.e., 8% of features have frames more than 600). The average EER reductions are approximately 35.6%, 43.8%, and 18.2% when M=200, M=400, and M=600, respectively. It is reasonable to infer that the higher the α, the higher the efficiency of the proposed method on reducing EER. Therefore, we recommend considering α to find a proper value of M when using the proposed method.

Next, we discuss the results in terms of the combination method. Contrary to our prediction that some specific combination methods would achieve lower EERs than others, there is no consistent tendency in terms of the combination method. Instead, the optimal combination method (i.e., the methods that showed the EER underlined in Table 1) differed for each case. Furthermore, in many cases, there were mismatches in the optimal combination method between the development and evaluation trials, although the other conditions (e.g., model, M, and task) were the same. Although *Fusion* achieved the lowest EER in most cases or EERs close to the lowest, it is inefficient because it requires multiple networks with different combination methods. Therefore, it is encouraged to develop a new combination method to achieve stable performances (e.g., consistent tendency across trials/conditions), left for future research.

### 5.2. Ablation Study

Throughout this paper, we have claimed that the proposed method reduces detection errors by increasing the amount of information available at one time. In this section, we support this claim through additional experiments in the following three factors: time range, flipping, and randomness of target length. We set the segment length M to 400 for all the experiments.

In Section 3.2, we doubted whether the proposed method still can reduce the detection errors when two segments in a pair always cover the same time range. To solve this doubt, we first compared the EERs between the proposed systems and its variations where both segments in a pair always cover the same time range (denoted as *ST*). Table 2 illustrates the oracle EERs of systems with the proposed method and *ST* for the LA and PA tasks, respectively. We focus on the results in the evaluation trials. If the detection errors are reduced even with *ST*, as with the proposed method, we no longer claim that the mechanism of the proposed method reducing the errors is to increase the amount of information available at one time. Instead, it might be because flipping the frame order has an effect of data augmentation, which led to the error reduction.

For the LA task, the proposed method exhibited lower EERs than *ST*, approximately 3.2% on average (−30.3% to 33.5%). Compared with the baseline, the proposed method exhibited lower EERs, approximately 8.2% on average, where 23 out of 29 proposed systems (except *average*) achieved the lower EERs. *ST* exhibited lower EERs than the baseline, approximately 2.5% on average, where 14 out of 29 systems exhibited the lower EERs with *ST*. Contrary to our prediction, there was no dominant performance gap between the proposed method and *ST*, albeit these results are still in accordance with our claim on the mechanism of the proposed method.

Unlike in the LA task, *ST* does not helpfully reduce EER at all for the PA task. The proposed method exhibited considerably lower EERs than *ST* for all cases, except the system of the MobileNetV2 with *vmax*. The EERs of the proposed methods are significantly lower than those of *ST*, approximately 46% on average (−24.2% to 64.5%). Moreover, compared with the baseline, the 27 out of 29 systems exhibited lower EERs when using the proposed method, approximately 43.2% on average, whereas 13 out of 29 systems exhibited lower EERs when using *ST* than the baseline, approximately merely 6.3% on average.

To further analyze the effect of the flipping the frame order, we conducted additional experiments to compare the oracle EERs between the baseline, proposed, backward-only (denoted as *BO*), and data-augmented (denoted as *Augment*) systems. *BO* is a variant of the baseline where only the backward segments $B_{i}$ were used in both the training and evaluation phases, not the forward segments $F_{i}$. *Augment* is identical to the baseline, but the total quantity of data is exactly twice the baseline in both the training and evaluation phases. These doubled amounts of data correspond to all the forward and backward segments, obtained by bidirectional feature segmentation. *Augment* treats two segments in each pair as individual data, not receiving them at one time. Therefore, the baseline, *BO*, and *Augment* have the same amount of information available at one time.

Table 3 presents the oracle EERs of the baseline, proposed, *BO*, and *Augment* systems in the development and evaluation trials of the LA and PA tasks. For the proposed systems, we presented the average of the EERs obtained with each combination method (excluding *Fusion*).

For the evaluation trials of the LA task, both *BO* and *Augment* exhibited higher EERs than the proposed systems on average, higher by approximately 14.9% (−9.4% to 40.5%) and 10% (−13.3% to 53.1%), respectively. Compared to the baseline systems, *BO* exhibited higher EERs, approximately 5.2% (−26.3% to 23%) on average. *Augment* exhibited almost the same EERs on average as the baseline, lower by only approximately 0.01% (−27.7 to 23.6%).

For the PA task, the proposed method always exhibited the lowest EERs by large margins. In the development trials, *BO* exhibited higher EERs on average, by approximately 11.5% (−18.5% to 57.7%) and 138.8% (32% to 228.8%) than the baseline and proposed systems, respectively. *Augment* exhibited higher EERs on average, by approximately 12.9% (−13.9% to 62.3%) and 141.7% (29.3% to 218%) than the baseline and proposed systems, respectively. In the evaluation trials, *BO* exhibited higher EERs on average, by approximately 3.6% (−19.2% to 40.5%) and 89% (14.5% to 138.6%) than the baseline and proposed systems, respectively. *Augment* exhibited higher EERs on average, by approximately 35.2% (17.2% to 94.7%) and 145.1% (63.7% to 199.1%) than the baseline and proposed systems, respectively.

As presented in Table 2 and Table 3, it can be observed that the proposed method generally exhibited lower EERs than *ST*, *BO*, and *Augment* in both the LA and PA tasks. In the LA task, the average performance gains with *ST*, *BO*, *Augment*, and the proposed method compared to the baseline are 2.5%, −5.2%, 0.01%, and 8.2%, respectively. In the evaluation trials of the PA task, the average performance gains with *ST*, *BO*, *Augment*, and the proposed method compared to the baseline are 6.3%, −3.6%, −35.2%, and 46%, respectively. These results support the claim that it is important to increase the amount of available information at one time for robust spoofing detection. The proposed method can satisfy this requirement, which is difficult, by merely flipping the frame order.

The next experiment compares the EERs of *2ch* and *2ch_s*, introduced in Section 3.2.4. Table 4 presents the oracle EERs of systems on the development and evaluation trials of the LA and PA tasks. The x-vector network was excluded in this experiment because it cannot receive a multi-channel input, as explained in Section 3.2.3.

For the LA task, *2ch_s* exhibited higher EERs than *2ch* in the evaluation trials, by approximately 10.8% on average (−11.2% to 32.4%). Three of the five models exhibited lower EERs with *2ch* than with *2ch_s*.

For the PA task, *2ch_s* exhibited distinctly higher EERs than *2ch*, regardless of the model. It exhibited approximately 134.1% (66.8% to 225.1%) and 79.7% (59.6% to 99.3%) higher EERs in the development and evaluation trials, respectively. *2ch* exhibited lower EERs than *2ch_s*. Consistent with our expectation, this is due to the additional parameters that provide different modeling from the existing parameters at the input feature level.

The last experiments were conducted to demonstrate that the proposed method is better than on-the-fly data loader on spoofing detection, as introduced in Section 1. Table 5 illustrates the oracle EERs of the baseline, on-the-fly data loader (denoted as *OTF*), and proposed systems in the development and evaluation trials of the LA and PA tasks. For the on-the-fly data loader, we sampled the target length for each iteration from a uniform distribution on the interval [200, 600] to match the average number of frames that CNNs take at one time with the baseline.

For the evaluation trials of the LA task, the proposed systems showed relatively 10.3% (−3.4% to 28.9%) lower EERs than *OTF* on average. Compared to the baseline systems, *OTF* exhibited higher EERs, approximately 4.3% (−21.3% to 39.7%) on average, whereas the proposed systems exhibited lower EERs, approximately 8.2% (−5.7% to 18.6%), than the baseline systems on average.

For the PA task, the proposed method showed relatively 58.6% (35.6% to 69%) and 51.8% (38.4% to 67.5%) lower EERs than *OTF* on average in the development and evaluation trials, respectively. Compared to the baseline systems, *OTF* exhibited higher EERs, approximately 20.4% (−4.1% to 59.3%) and 19.2% (−2.3% to 42.4%) on average, in the development and evaluation trials, respectively. In contrast, the proposed systems showed significantly lower EERs, approximately 50.9% (30.9% to 63.3%) and 41.8% (15% to 60.4%) on average, in the development and evaluation trials, respectively.

## 6. Conclusions

We proposed a bi-point input method based on bidirectional feature segmentation. It enables CNNs that receive variable-length inputs to increase the amount of available information at one time. Each feature is decomposed into two sets of segments (i.e., forward and backward segment sets) along two directions (i.e., positive and negative time directions). Then, the CNN receives a pair of two segments (i.e., one forward segment and one backward segment) as an input at one time. We also evaluated various combination methods. The results from both segments are combined using one of the proposed combination methods before being fed into the final classifier to aggregate the information from both segments. The proposed method achieved lower EERs than the conventional method on both the LA and PA tasks of ASVspoof 2019, while minimizing changes in the network structure. Based on these results, we claim that the proposed method with suitable combination methods can function as robust spoofing detection systems by increasing the amount of information available at one time. We also provided a rough guidance to set a proper segment length without evaluation.

In the future, we will extend the proposed method to receive more than two segments at one time, for a further reduction in detection errors. To achieve a stable performance without the fusion, we will investigate how to aggregate the information in all the pairs of segments more efficiently. Moreover, we will apply our proposed method to different tasks, such as speaker verification, acoustic scene classification, and others that use different types of time-series input other than audio. We predict that the proposed method would also benefit tasks where the discriminative attributes between the classes are spread across the entire time domain, like spoofing detection.

## Figures and Tables

**Figure 1 sensors-22-04483-f001:**
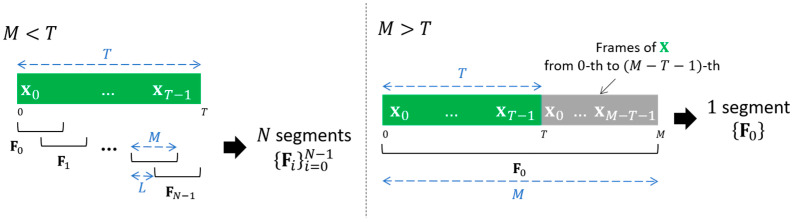
The overview of the conventional feature segmentation when (**left**) M<T and (**right**) M>T, where $X=\left[x_{0},\ldots ,x_{T-1}\right]$ is a feature with the length of T, M is the segment length, L is the shift interval for segmentation, and $F_{i}$ is the i -th segment.

**Figure 2 sensors-22-04483-f002:**
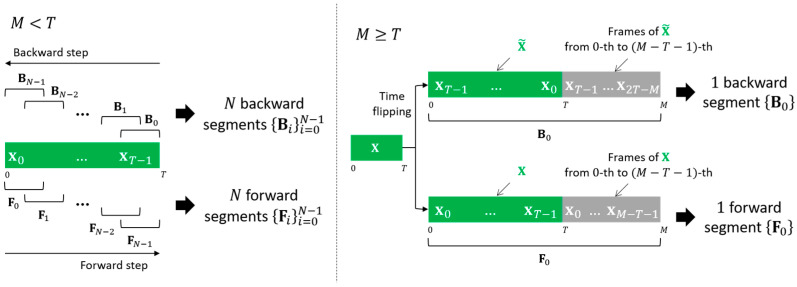
The overview of the proposed bidirectional feature segmentation when (**left**) M<T and (**right**) M>T, where $X=\left[x_{0},\ldots ,x_{T-1}\right]$ is a feature with the length of T, M is the segment length, and $F_{i}$ and $B_{i}$ are the i -th forward and backward segments, respectively.

**Figure 3 sensors-22-04483-f003:**
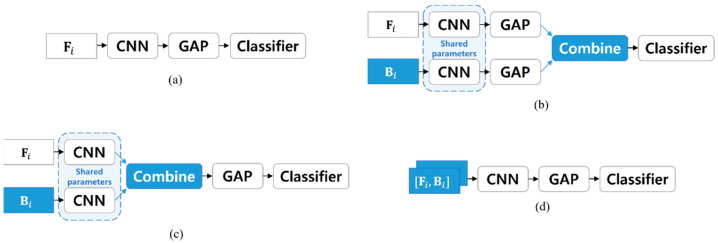
The framework of (**a**) conventional and (**b**–**d**) proposed systems, where GAP stands for global average pooling, Combine (**b**,**c**) stands for one of the appropriate combination methods (described in Section 3.2.1 and Section 3.2.2, respectively), and $F_{i}$ and $B_{i}$ are the i -th forward and backward segments, respectively. The proposed systems consist of (**b**) embedding-level combination, (**c**) feature-map level combination, and (**d**) two-channel input. The modules for the proposed method are marked in blue.

**Figure 4 sensors-22-04483-f004:**
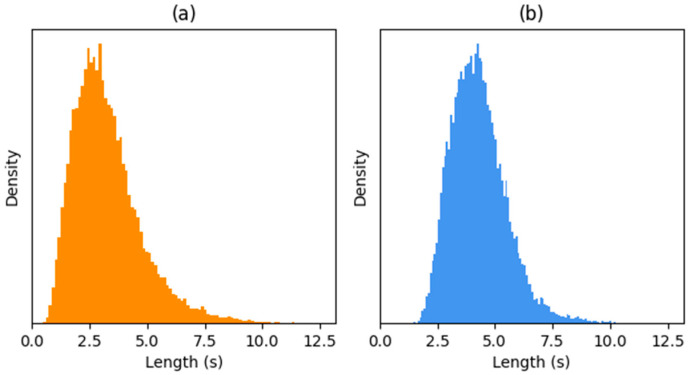
The histogram of the utterance lengths in ASVspoof (**a**) LA and (**b**) PA scenarios.

**Table 1 sensors-22-04483-t001:** EERs (%) of systems on the development and evaluation trials of ASVspoof 2019 LA and PA scenarios. For each model, the lowest EER is highlighted in bold (comparison including *Fusion*) or is underlined (comparison excluding *Fusion*).

Model	Method	LA						PA					
		M=200		M=400		M=600		M=200		M=400		M=600	
		Dev.	Eval.	Dev.	Eval.	Dev.	Eval.	Dev.	Eval.	Dev.	Eval.	Dev.	Eval.
SE-ResNet	Baseline	0	14.099	0	10.388	0	12.551	2.152	3.197	0.811	1.255	0.519	0.985
	*concat*	0	15.486	0	11.148	0	** 9.410 **	1.261	2.641	0.370	0.945	0.612	1.007
	*vmax*	0	13.635	0	12.157	0	10.633	1.279	2.118	0.704	1.327	0.998	1.975
	*vmean*	0	13.637	0	9.081	0	12.211	1.573	3.300	0.593	1.337	0.926	1.702
	*fmax*	0	15.812	0	9.868	0	13.353	1.388	2.875	0.534	1.105	0.484	0.934
	*2ch*	0	13.923	0	11.434	0	12.498	2.538	4.378	0.597	1.063	0.316	0.912
	Fusion	0	**13.131**	0	**9.001**	0	10.236	**0.737**	**1.940**	**0.261**	**0.653**	**0.168**	**0.614**
X-vector Network (TDNN)	Baseline	0.009	17.321	0	13.395	0	6.947	3.965	8.297	1.797	3.538	0.908	1.664
	*concat*	0.009	15.690	0	** 7.900 **	0	**6.718**	1.762	3.964	0.667	1.520	0.834	1.366
	*vmax*	**0**	14.750	0.002	8.171	0	8.811	1.296	3.936	0.721	1.072	0.721	1.056
	*vmean*	0.082	16.558	0	11.081	0	7.763	2.170	3.897	0.797	1.708	0.856	1.520
	*statc*	0.040	16.927	0	8.620	0.002	12.143	2.037	3.892	0.926	1.752	0.797	1.156
	Fusion	0.004	**14.451**	0	8.389	0	8.552	**0.628**	**2.205**	**0.444**	**0.950**	**0.610**	**0.902**
DenseNet	Baseline	0	16.232	0	10.130	0	9.818	2.296	3.155	0.850	1.360	0.337	0.713
	*concat*	0	12.120	0	** 7.139 **	0	11.405	0.854	1.593	0.353	0.658	0.296	0.580
	*vmax*	0	11.109	0	10.744	0	** 8.743 **	0.700	1.116	0.366	0.857	0.222	0.492
	*vmean*	0	12.509	0	12.628	0	12.158	1.090	1.708	0.296	0.564	0.238	0.476
	*fmax*	0	14.279	0	14.535	0	11.828	0.630	1.609	0.333	0.845	0.242	0.508
	*2ch*	0	13.963	0	9.258	0	11.951	2.164	2.798	0.370	0.956	0.226	0.542
	Fusion	0	**11.027**	0	7.410	0	10.023	**0.444**	**0.885**	**0.148**	**0.425**	**0.129**	**0.303**
MobileNetV2	Baseline	0	19.511	0	11.299	0	8.292	2.667	4.666	1.076	2.233	0.409	0.785
	*concat*	0	13.584	0	8.307	0	10.414	1.649	3.466	0.667	1.597	0.386	0.878
	*vmax*	0	12.794	0	10.223	0	8.767	1.333	3.217	1.022	2.206	0.279	0.872
	*vmean*	0	12.387	0	8.280	0	10.416	1.185	2.996	0.799	1.873	0.501	1.039
	*fmax*	0	13.433	0	9.950	0	8.768	1.076	3.228	0.756	1.592	0.388	0.928
	*2ch*	0	14.167	0	10.727	0	8.537	1.889	4.533	** 0.337 **	** 0.962 **	0.279	0.901
	Fusion	0	**10.809**	0	**7.815**	0	8.363	**0.741**	**2.387**	0.407	0.978	**0.185**	**0.520**
ShuffleNetV2	Baseline	0	20.479	0	11.748	0	11.051	4.170	4.374	1.392	2.255	0.904	1.382
	*concat*	0	** 13.664 **	0	9.433	0	8.849	1.240	2.161	0.462	0.917	0.388	0.867
	*vmax*	0	19.443	0	10.265	0	10.007	1.146	2.102	0.503	0.823	0.388	0.746
	*vmean*	0	17.975	0	9.705	0	9.912	1.390	2.366	0.335	0.869	0.353	0.696
	*fmax*	0	18.204	0	9.571	0	10.048	1.094	2.123	0.514	1.128	0.412	0.779
	*2ch*	0.002	19.798	0	10.472	0.038	9.764	2.168	3.737	0.739	1.260	0.593	0.951
	Fusion	0	13.923	0	**8.744**	0	**8.620**	**0.760**	**1.581**	**0.279**	**0.603**	**0.207**	**0.492**
MNASNet	Baseline	0	19.050	0	10.608	0	11.205	2.263	5.052	0.834	2.232	0.353	0.995
	*concat*	0	** 11.652 **	0	10.249	0	11.624	1.207	3.570	0.255	0.912	0.370	0.614
	*vmax*	0	17.362	0	11.895	0	11.107	1.630	4.416	0.353	1.156	0.168	0.718
	*vmean*	0	14.333	0	7.207	0	** 8.863 **	1.316	4.388	0.279	1.321	0.164	0.685
	*fmax*	0	15.894	0	7.570	0	11.327	1.037	3.642	0.316	0.969	0.152	0.615
	*2ch*	0	17.933	0	9.341	0	10.331	1.540	3.687	0.409	1.238	0.240	0.857
	Fusion	0	13.705	0	**7.029**	0	9.463	**0.663**	**2.732**	**0.148**	**0.636**	**0.094**	**0.359**

**Table 2 sensors-22-04483-t002:** Oracle EERs (%) of systems on the development and evaluation trials of ASVspoof 2019 LA and PA scenarios, with the proposed and *ST* methods. For each model, the lowest EER is highlighted in bold.

Model	Method	LA				PA			
		Proposed		*ST*		Proposed		*ST*	
		Dev.	Eval.	Dev.	Eval.	Dev.	Eval.	Dev.	Eval.
SE-ResNet	*concat*	0	**7.967**	0	8.484	**0.370**	**0.708**	1.329	1.835
	*vmax*	0	9.517	0	**8.989**	**0.704**	**1.327**	0.983	1.520
	*vmean*	0	9.081	0	**9.012**	**0.593**	**1.156**	0.945	1.414
	*fmax*	0	**7.576**	0	8.893	**0.534**	**1.105**	1.238	1.913
	*2ch*	0	9.261	0	**7.941**	**0.597**	**1.040**	1.403	1.896
	average	0	8.680	0	**8.664**	**0.560**	**1.067**	1.180	1.716
X-vector Network (TDNN)	*concat*	0	**6.352**	0	8.267	**0.667**	**1.493**	1.760	3.108
	*vmax*	**0.002**	**6.741**	0.004	8.674	**0.721**	**1.027**	1.630	2.665
	*vmean*	0	**7.804**	0	10.524	**0.797**	**1.497**	1.817	3.814
	*statc*	0	**6.309**	0	9.488	**0.926**	**1.581**	1.630	2.924
	average	0.001	**6.802**	0.001	9.238	**0.778**	**1.400**	1.709	3.128
DenseNet	*concat*	0	6.868	0	**6.472**	**0.353**	**0.542**	0.908	1.194
	*vmax*	0	6.717	0	**6.499**	**0.366**	**0.719**	1.133	1.414
	*vmean*	0	7.981	0	**7.108**	**0.296**	**0.564**	1.148	1.587
	*fmax*	0	7.628	0	**5.944**	**0.333**	**0.702**	0.908	1.332
	*2ch*	0	7.828	0	**7.548**	**0.370**	**0.796**	0.737	0.961
	average	0	7.404	0	**6.714**	**0.344**	**0.665**	0.967	1.298
MobileNetV2	*concat*	0	**6.067**	0	6.717	**0.667**	**1.398**	1.004	2.034
	*vmax*	0	7.439	0	**5.710**	1.022	2.150	**0.595**	**1.731**
	*vmean*	0	**7.533**	0	8.239	**0.799**	**1.847**	1.037	2.311
	*fmax*	0	6.186	0	**6.172**	**0.756**	**1.371**	0.815	1.979
	*2ch*	0	**6.242**	0	7.738	**0.337**	**0.935**	0.980	2.217
	average	0	**6.693**	0	6.915	**0.716**	**1.540**	0.886	2.054
ShuffleNetV2	*concat*	0	**6.609**	0	9.166	**0.462**	**0.885**	1.649	1.924
	*vmax*	0	7.032	0	**6.564**	**0.503**	**0.823**	1.630	2.299
	*vmean*	0	**6.405**	0	7.136	**0.335**	**0.869**	1.353	2.278
	*fmax*	0	7.454	0	**7.330**	**0.514**	**1.061**	1.294	1.869
	*2ch*	0	8.786	0	**8.144**	**0.739**	**1.227**	1.702	2.846
	average	0	**7.257**	0	7.668	**0.511**	**0.973**	1.526	2.243
MNASNet	*concat*	0	6.080	0	6.079	**0.255**	**0.774**	0.908	1.974
	*vmax*	0	**5.398**	0	5.452	**0.353**	**0.818**	0.651	1.951
	*vmean*	0	**5.384**	0	7.098	**0.279**	**0.923**	0.963	2.427
	*fmax*	0	4.840	0	**4.756**	**0.316**	**0.846**	0.776	2.095
	*2ch*	0	6.840	0	**6.349**	**0.409**	**1.039**	0.926	2.869
	average	0	**5.708**	0	5.947	**0.322**	**0.880**	0.845	2.263

**Table 3 sensors-22-04483-t003:** Oracle EERs (%) of systems in the development and evaluation trials of ASVspoof 2019 LA and PA, with four different methods (i.e., baseline, *BO*, *Augment*, and proposed). For each model, the lowest EER is highlighted in bold.

Model	Method	LA		PA	
		Dev.	Eval.	Dev.	Eval.
SE-ResNet	Baseline	0	8.211	0.811	1.255
	*BO*	0	8.347	1.279	1.763
	*Augment*	0	**7.940**	1.316	2.443
	Proposed	0	8.680	**0.560**	**1.067**
X-vector Network (TDNN)	Baseline	0	8.157	1.797	3.538
	*BO*	0	9.557	1.464	2.858
	*Augment*	0	10.415	1.595	4.145
	Proposed	0	**6.802**	**0.778**	**1.400**
DenseNet	Baseline	0	9.095	0.850	1.360
	*BO*	0	**6.705**	1.131	1.581
	*Augment*	0	6.947	1.094	1.713
	Proposed	0	7.404	**0.344**	**0.776**
MobileNetV2	Baseline	0	7.695	1.076	1.885
	*BO*	0	7.791	0.945	1.763
	*Augment*	0	7.043	0.926	2.521
	Proposed	0	**6.693**	**0.716**	**1.540**
ShuffleNetV2	Baseline	0	7.303	1.392	2.102
	*BO*	0	8.985	1.514	2.024
	*Augment*	0	**6.293**	1.499	2.549
	Proposed	0	7.257	**0.511**	**0.973**
MNASNet	Baseline	0	6.092	0.834	2.224
	*BO*	0	6.960	0.834	2.100
	*Augment*	0	7.400	0.867	2.632
	Proposed	0	**5.708**	**0.322**	**0.880**

**Table 4 sensors-22-04483-t004:** Oracle EERs (%) of systems (excluding the x-vector network) in the development and evaluation trials of ASVspoof 2019 LA and PA, with the methods of *2ch* and *2ch_s*. For each model, the lowest EER is highlighted in bold.

Model	Method	LA		PA	
		Dev.	Eval.	Dev.	Eval.
SE-ResNet	*2ch*	0	9.261	**0.597**	**1.040**
	*2ch_s*	0	**8.226**	0.996	1.731
DenseNet	*2ch*	0	**7.828**	**0.370**	**0.796**
	*2ch_s*	0	9.257	1.203	1.527
MobileNetV2	*2ch*	0	**6.242**	**0.337**	**0.935**
	*2ch_s*	0	8.267	0.869	1.863
ShuffleNetV2	*2ch*	0	8.786	**0.739**	**1.227**
	*2ch_s*	0	**7.775**	1.538	1.958
MNASNet	*2ch*	0	**6.840**	**0.409**	**1.039**
	*2ch_s*	0	8.620	0.869	1.885

**Table 5 sensors-22-04483-t005:** Oracle EERs (%) of systems in the development and evaluation trials of ASVspoof 2019 LA and PA, with three different methods (i.e., baseline, *OTF*, and proposed). For each model, the lowest EER is highlighted in bold.

Model	Method	LA		PA	
		Dev.	Eval.	Dev.	Eval.
SE-ResNet	Baseline	0	**8.211**	0.811	1.255
	*OTF*	0	9.626	1.292	1.787
	Proposed	0	8.680	**0.560**	**1.067**
X-vector Network (TDNN)	Baseline	**0**	8.157	1.797	3.538
	*OTF*	0.002	8.158	1.723	3.455
	Proposed	**0**	**6.802**	**0.778**	**1.400**
DenseNet	Baseline	0	9.095	0.850	1.360
	*OTF*	0	**7.162**	1.111	1.393
	Proposed	0	7.404	**0.344**	**0.776**
MobileNetV2	Baseline	0	7.695	1.076	1.885
	*OTF*	0	7.163	1.111	2.498
	Proposed	0	**6.693**	**0.716**	**1.540**
ShuffleNetV2	Baseline	0	7.303	1.392	2.102
	*OTF*	0	10.200	1.589	2.493
	Proposed	0	**7.257**	**0.511**	**0.973**
MNASNet	Baseline	0	6.092	0.834	2.224
	*OTF*	0	5.914	0.996	2.709
	Proposed	0	**5.708**	**0.322**	**0.880**

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
