# Peer review of "BPCNN: Bi-Point Input for Convolutional Neural Networks in Speaker Spoofing Detection"

_sensors, 2022, doi:10.3390/s22124483_

Round 1
Reviewer 1 Report
This manuscript sensors-1745579 proposed a convolutional neural networks (CNNs) that can handle variable-length input features (e.g., speech utterances). Feeding input features into a CNN in a mini-batch unit requires that all features in each mini-batch have the same shape. A set of variable-length features cannot be directly fed into a CNN because they commonly have different lengths. Feature segmentation is a dominant method for CNNs to handle variable-length features, where each feature is decomposed into fixed-length segments. A CNN receives one segment as an input at one time. The experimental results reveal that the proposed method reduces the relative equal error rate (EER) by approximately 17.2% and 43.8% on average for the logical access (LA) and physical access (PA) tasks, respectively. I feel the experiment results are sufficient. It was a pleasure reviewing this work and I can recommend it for publication in Sensors after a major revision. I respectfully refer the authors to my comments below.
1) The English needs to be revised throughout. The authors should pay attention to the spelling and grammar throughout this work. I would only respectfully recommend that the authors perform this revision or seek the help of someone who can aid the authors. For example,
--(Line 181) “3. The Proposed Method”->”The Proposed Method”.
2) (Page 2) The references in this sentence is not accuracy, such as “Accordingly, many cutting-edge spoofing detection systems are based on CNNs [7-26]”. There are too many one-time references.
3) (Section I, Introduction, Paragraph I) The reviewer suggest to add some related work in the original statement “These locally connected information can be effectively modeled by convolutional neural networks (CNNs) [1][2][3]”.{<1> CARM: Confidence-aware recommender model via review representation learning and historical rating behavior in the online platforms <2> Anisotropic angle distribution learning for head pose estimation and attention understanding in human-computer interaction; <3> NGDNet: Nonuniform Gaussian-label distribution learning for infrared head pose estimation and on-task behavior understanding in the classroom}
4) (Line 138) The original figure 1 is not clear. Please redraw this figure clearly.
5). The reviewer hopes the introduction section in this paper can introduce more studies in recent years. The reviewer suggests authors don't list a lot of related tasks directly. It is better to select some representative and related literature or models to introduce with certain logic. For example, the latter model is an improvement on one aspect of the former model.
6) In the Introduction part, “main contributions” is best to list clearly by breaking it down into three points.
7) (Page 8, Line 272) The reviewer suggests authors introduce clearly the overall flow of Figure 3 (in the body or in the picture description).
8) (Page 3, Line 57. Reference Target) I suggest two references to this sentence that “Like other types of neural networks [1][2], a CNN is also trained in a mini-batch unit.”{<1>DOI: 10.1109/TII.2021.3128240; <2> DOI: 10.1109/TII.2022.3143605}. The references are about the neural networks.
9) Experimental pictures or tables should be described and the results should be analyzed in the picture description so that readers can clearly know the meaning without looking at the body. For example, describe the orange and blue curves in Figure 4 and describe the results of the analysis of this phenomenon.
10) All the best values are marked by red font. Please check this format. The reviewer suggests to revise as the BOLD font.
11). (Page 3, Section 2-Conventional Feature Segmentation, the Paragraph I) The original statement is suggested to revised as “Feature segmentation is a method that decomposes all the variable-length features into fixed-length segments along the time axis. With the deep learning technology successful using in computer vision [1][2] ([1] DOI: 10.1109/TMM.2021.3081873, [2] DOI: 10.1109/TII.2019.2930463)) and natural language processing [3][4] (<3>DOI: 10.1109/TNNLS.2021.3055147, <4> Recalibration Convolutional Networks for Learning Interaction Knowledge Graph Embedding), it also been employed in the deep learning-based Feature Segmentation.”
12). Please give an abbreviation name for your proposed model, and explain the advantage of bi-point input method, comparing with other type input in automatic speaker verification task.
My overall impression of this manuscript is that it is in general well-organized. The work seems interesting and the technical contributions are solid. I would like to check the revised manuscript again.
Reviewer 2 Report
Paper is an extension of [13] (Yoon S.-H.; Yu, H.-J. Multiple points input for convolutional neural networks in replay attack detection. In Proceedings of IEEE International Conference on Acoustics, Speech, and Signal Processing (ICASSP), May 2020, pp. 6444-6448) of the same authors.
It presents a system based on multi-points input CNN in order to detect various spoofing attacks in Automatic speaker verification.
The major drawback of the paper is the lack of performance comparisons versus other approaches. Only feature segmentation is used as baseline but with the same CNN approach. In my opinion, the section "Experiments" should contain comparisons versus recent approaches which do not use CNNs. As claimed by the authors, some challenges have been organized periodically by the researchers in this field. Authors could add as baselines one or more approaches with best results obtained in these researches using the same dataset.
Some minor issues:
- row 135: the sentence
"until the (iL + M)-th frame xiL+M , Fi = X[iL ∶ iL + M] = [xiL , ... , x iL+M−1 ], where iL + M ≤ T for all i"
in my opinion is wrong, it should be updated as
"until the (iL + M)-th frame xiL+M-1 , Fi = X[iL ∶ iL + M-1] = [xiL , ... , x iL+M−1 ], where iL + M - 1 ≤ T for all i"
- row 280: authors should explain better how global average pooling (GAP) is applied;
- rows 288-292: authors should explain how "vmax" and "vmean" are applied on vectors?
